# Low-Frequency Resonant Magnetoelectric Effect in a Piezopolymer-Magnetoactive Elastomer Layered Structure at Different Magnetization Geometries

**DOI:** 10.3390/polym16070928

**Published:** 2024-03-28

**Authors:** Dmitrii V. Savelev, Dmitri A. Burdin, Leonid Y. Fetisov, Yuri K. Fetisov, Nikolai S. Perov, Liudmila A. Makarova

**Affiliations:** 1Research and Educational Center “Magnetoelectric Materials and Devices”, MIREA–Russian Technological University, 119454 Moscow, Russia; dimsav94@gmail.com (D.V.S.); phantastic@mail.ru (D.A.B.); fetisovl@yandex.ru (L.Y.F.); fetisov@mirea.ru (Y.K.F.); 2Faculty of Physics, Lomonosov Moscow State University, 119991 Moscow, Russia; perov@magn.ru; 3Institute of Physics, Mathematics & IT, Immanuel Kant Baltic Federal University, 236041 Kaliningrad, Russia

**Keywords:** magnetoactive elastomer, magnetoelectric effect, multiferroic, iron particles, piezopolymer, layered structure, bending deformation, frequency tuning

## Abstract

The search for novel materials with enhanced characteristics for the advancement of flexible electronic devices and energy harvesting devices is currently a significant concern. Multiferroics are a prominent example of energy conversion materials. The magnetoelectric conversion in a flexible composite based on a piezopolymer layer and a magnetic elastomer layer was investigated. The study focused on investigating the dynamic magnetoelectric effect in various configurations of external alternating and constant homogeneous magnetic fields (L-T and T-T configurations). The T-T geometry exhibited a two orders of magnitude higher coefficient of the magnetoelectric effect compared to the L-T geometry. Mechanisms of structure bending in both geometries were proposed and discussed. A theory was put forward to explain the change in the resonance frequency in a uniform external field. A giant value of frequency tuning in a magnetic field of up to 362% was demonstrated; one of the highest values of the magnetoelectric effect yet recorded in polymer multiferroics was observed, reaching up to 134.3 V/(Oe∙cm).

## 1. Introduction

Magnetoelectric structures belong to smart materials and have a wide range of applications from sensors to magnetic memory devices. These materials contain both ferromagnetic and ferroelectric components, which can be organized into sub-phases or sub-systems. Alternatively, these features can be observed in single-phase multiferroics. These materials demonstrate magnetoelectric behavior, namely the ability to change polarization with the magnetic field, and, vice versa, the ability to change magnetization with the electric field. Magnetometers, frequency tunable inductors and filters, antennas, and energy harvesters, etc., are the examples of all kinds of technological applications for magnetoelectric materials [1,2,3,4,5,6].

The value of the magnetoelectric (ME) effect in composite multiferroics depends both on the properties of components and interface coupling. It is characterized by ME coefficient *α*, which is equal to the ratio between the electric field *δE* generated in the piezoelectric (PE) component and the amplitude of magnetic field *δH*: *α* = *δE/δH* [7,8]. The direct ME effect in composite structures originates owing to magnetostriction in the ferromagnetic component, which manifests as the deformation under an external magnetic field. Then this deformation is transferred to the piezoelectric component, in which the electric voltage appears due to the piezoelectric effect. Traditional examples of ferromagnetic materials with high magnetostriction coefficients are metals (Ni, Fe), Terfenol-D and other rare earth alloys, amorphous magnetic alloys; examples of piezoelectric materials are lead based ceramics (PZT, PMN-PT), and piezoelectric single crystals [1,2,9,10,11]. However, the flexibility, stretchability, and foldability are limited for these materials, which can impair the effectiveness and service life of entirely magnetoelectric materials [1,2,12].

Flexible magnetoelectric elements are based on polymer components that allow their incorporation, to some extent, into technological device applications, such as flexible displays, soft tools, magnetoactive robotics, and biomedical applications [12,13,14,15,16,17]. The most commonly used polymer component for multiferroics is piezoelectric polymer poly(vinylidene fluoride), PVDF. The electroactive state of the PVDF can be realized in β-phase, where the polarization leads to electrical voltage production via mechanical deformation [16]. The layered structures and the composites where the ferromagnetic particles embedded inside polymer PVDF are often employed for investigations of the ME effect. The ME coefficient in such structures can reach significant values of several V/(cm∙Oe) [15,18,19,20,21]. These composites can be prepared for different shapes and geometries and can be biocompatible. One of the highest values of the ME coefficient was established for a PVDF–TrFE/Metglas laminated structure (850 V/(cm∙Oe)) [21] that can be applied in magnetic sensing and energy harvesting devices [22,23,24,25].

The most pronounced magnetoelectric effect is exhibited at a resonance frequency. Depending on the different applications, the resonant frequency is of interest [1,2,22,23,24,25,26]. Its value depends on the dimensions of the composite structure as much as on its mechanical properties [27]. Flexible structures have low values of Young’s moduli, which makes it feasible to reduce their frequencies of bending vibrations to tens of Hz—units of kHz [21,22,23,24,25,26]. This frequency range is promising for the design of biomedical devices, energy harvesters, and magnetic field sensors [22,23,24,25,26]. To the best of our knowledge, the lowest resonance frequency of 27.8 Hz was recorded for the PVDF–Metglas structure [21]. To minimize the resonance frequency, materials with a low Young’s modulus can be used. Among such materials are magnetoactive elastomers, MAEs, which consist of an electrically neutral polymer soft matrix with embedded iron microparticles [28], where Young’s moduli is about 10^3^–10^6^ Pa, which is several orders of magnitude smaller than for PVDF [1,2,12]. 

A two-layered structure composed of PVDF and magnetic elastomer was investigated earlier [29,30,31,32,33]. The initial experiments of the magnetoelectric effect were carried out in a pulsed magnetic field of different forms oriented transversally [29,30,31]. Such a configuration is also called T-T [7]. Under various forms of magnetic field treatment, the ferromagnetic microparticles are forced from the field along the field lines. Due to elastic polymer matrix the MAE layer bends with the piezopolymer layer in the cantilever configuration, i.e., the structure is fixed from one edge. It was proven that the higher the mass fraction of magnetic particles as long as the thickness of MAE layer, the larger the ME effect that could be obtained [29,30]. In one work [31], it was found that value sinusoidal excitation is more effective than triangular. The authors of articles [29,30,31] suppose that the ME effect originates owing to a gradient in the magnetic field inside the MAE layer.

Resonance enhancement of the ME effect was also observed [32,33]. In a work [33], it was shown that the resonance frequency in the T-T configuration in the AC gradient magnetic field for the first resonance mode decreases with the increasing magnetic field. Similar results were obtained in another work [29]. It should be also noted that the generation of ME voltage harmonics can also be suggested [33]. In another work [32], measurements of the ME effect were carried out in a uniform magnetic field directed in the plane of the structure (so-called L-T geometry). The ME effect emerges in such a case due to the magnetodeformation of the MAE layer. It was also established that there is optimal thickness *t*_MAE_ ≈ 1.9 mm, when the ME coefficient reaches its maximal value.

It was previously shown that PVDF–MAE structures exhibit remarkable frequency tuning under the action of a magnetic field. Its greatest value in a field of 3 kOe was 216% [32]. The resonant frequency tuning usually reaches some tenths of percentages. The maximum values were attained for layered structures: 4.4% for the piezoceramic-based composite PZT–hematite and 24% for the piezopolymer-based composite PVDF–hematite in a saturated state [34]. The value of relative frequency change is commonly calculated between minimum- and maximum-saturated values of resonant frequency in the field dependency of the resonant frequency. This allows us to compare frequency tuning values in different magnetic field ranges. The study of the controlled resonant frequency tuning via the ME effect across a wide frequency range in flexible composite structures is of significant interest for subsequent practical applications, especially for ME tunable low-frequency antennas and sensors [26,35,36].

The work presents the results of the resonance magnetoelectric effect analysis in a bilayer structure “piezopolymer PVDF–magnetic elastomer” (PEP-MAE-Fe77) with a 3 mm MAE layer thickness, which contains iron microparticles with a mass fraction of 77 mass %. Young’s modulus of the MAE composite was 90 kPa, which allowed us to investigate the magnetoelectric effect at low frequencies. The AC and DC homogeneous magnetic fields were applied simultaneously in two cases: the L-T configuration and T-T configuration. In the first case, the influence of gravity was eliminated, and the ME effect was associated with the magnetodeformational effect in the magnetic elastomer. In the second case the external magnetic field was applied perpendicular to the sample plane, and the electrical voltage was measured in the direction perpendicular to the plane of the PVDF. The mechanism of magnetoelectric conversion is associated with the ponderomotive force, which cause the rotation of MAE. The existence of such a force and the corresponding rotation have been described in other works [37,38]. The authors showed that for MAE, the membrane deflection in a uniform magnetic field occurred in a threshold manner at a finite magnitude of the applied field due to soft instability. Thus, the mechanisms of two ME effects are compared in this work, and a theory based on cantilever oscillation is proposed. The analytical and experimental results are in agreement. The maximum value of the ME effect obtained was 134.3 V/(cm∙Oe), which is one of the highest values among two-layered PVDF-based multiferroics. The giant frequency shift of ~362% was observed in the bias magnetic field and it is the largest value among existing results. Such a flexible magnetoelectric element allows to construct flexible technological devices without leakage currents, with remote access and good stretchability and foldability.

## 2. Materials and Methods

### 2.1. Composite Structure

The studies of the ME effect were carried out in the bilayer bending structure piezoelectric polymer–magnetic elastomer (PEP-MAE). The piezoelectric layer was a commercially available piezoelectric element PVDF (LTD0-028K, TE Instruments, Delft, The Netherlands) with conductive plates and protective covers. The thickness of the PVDF layer was *t*_PE_ = 28 µm and the plane sizes were 25 × 13 mm^2^ [32]. The magnetostrictive layer was a magnetoactive elastomer based on the silicone matrix SIEL (GNIIChTEOS, Moscow, Russia) [28,29] with carbonyl iron particles with an average size of 5 µm evenly distributed inside the matrix. The mass fraction of the particles was *ω*_Fe_ ≈ 77 mass %. The Young’s modulus of the MAE sample in zero magnetic field was *E*_2_ = 90 kPa. The preparation method of the MAE sample was described in [29]. The layer with in-plane dimensions of 19.7 × 13 mm^2^ and a thickness of 3 mm was cut from the prepared MAE. The MAE layer was bonded to the negative side of the piezopolymer using a silicone adhesive. The thin layer of silicone sealant adhesive provided a mechanical bond between the layers and a strain transfer between them. A photograph of the PEP-MAE composite structure is shown in Figure 1.

The magnetic properties of the MAE sample have been previously measured and described in detail in [29]. The hysteresis loop of the MAE material with 77 mass % concentration of iron particles is presented here in order to use the magnetic parameters for modelling.

To measure the magnetoelectric effect, the structure was rigidly fixed from one edge to a massive base at a distance L = 19.7 mm, equal to the length of the MAE layer from the free end. The sample was placed in a permanent magnetic field of up to 3 kOe (bias field), which was generated by an electromagnet connected to a TDK Lambda GENH600-1.3 DC (TDK Lambda Corporation, Tokyo, Japan) power supply, and the sample was excited by an AC magnetic field with a frequency *f* in the range from 10 to 200 Hz. Both fields were parallel to each other. Measurements were carried out for two orientations of the magnetic fields relative to the sample plane, which are described below.

The voltage generated by the structure was supplied to the SR560 preamplifier (Stanford Research Systems, Sunnyvale, CA, USA) with a gain of *K* = 1, operating in the differential input mode to avoid common-mode noise, and a high-pass filter with a cut-off frequency of *f_c-o_* = 1 Hz. The signal from the preamplifier was supplied to the input of an AKIP 2401 voltmeter (Shijiazhuang Suin Instruments Co., Shijiazhuang, Hebei, China) with an input resistance of 10 MΩ. The frequency dependence of the magnetoelectric voltage at different values of the DC magnetic field was measured to plot the dependence of the induced signal and resonance frequency on the bias field. The field strength was controlled using a LakeShore421 gaussmeter (Lake Shore Cryotronics, Westerville, OH, USA).

### 2.2. Geometry 1 (L-T)

In the first case, the sample clamped in a massive holder was put within the excitation coil, which formed an AC magnetic field with an amplitude of *h* = 7.6 Oe along its long side. The sample was positioned in a certain stand between the poles of the electromagnet, created bias magnetic field directed longitudinally in the samples plane. The schematic representation of the measurement configuration is illustrated in Figure 2a. The DC magnetic field was adjusted in steps of 0.1 kOe in the range from 0 to 2 kOe. The measuring setup was described in detail in [32].

### 2.3. Geometry 2 (T-T)

In the second case, the magnetic fields were directed perpendicular to the sample plane: the sample was put vertically using special holder between the poles of the electromagnet (see Figure 2b). AC magnetic field with an amplitude of up to 5.8 Oe was generated using electromagnetic coils fixed to the poles of the electromagnet. The measurement geometry is similar to that presented in Refs. [30,31].

## 3. Results

### 3.1. ME Effect in Geometry 1 (L-T)

The dependences of the induced voltage *u* in geometry 1 (L-T) on the frequency of the AC magnetic field *f* at different values of the DC magnetic field *H* are presented in Figure 3. Resonance peaks correspond to the first mode of bending oscillations of the structure. The resonance frequency *f*_1_ significantly shifts to higher frequencies with the increasing bias magnetic field. The voltage *u*_1_ increase in the frequency range of 200 Hz in the field of 0.5 kOe is due to the occurrence of the second mode of bending oscillations of the structure.

The dependences of the induced voltage *u*_1_ and resonant frequency *f*_1_ on the bias magnetic field *H* are shown in Figure 4.

The dependence of the induced voltage on the bias magnetic field has a classical form. Its shape is determined by the shape of the “piezomagnetic” coefficient, which is equal to the first derivative of the MAE layer magnetodeformation with respect to magnetic field [32]. The magnitude of the field in which the highest value of the ME voltage *u*_1_ ≈ 38 mV is observed is *H*_m1_ = 0.3 kOe, which is less than that obtained earlier in [32] for a structure with a similar thickness of the MAE layer. The maximum calculated MEE coefficient is *ά*_1_ = *u*_1_/(*t*_PE_*∙h*) *≈* 1.8 V/(cm∙Oe).

Figure 4b indicates that the resonant frequency grows monotonically with increasing magnetic field from the initial value of *f*_1-min_ ≈ 30.2 Hz in the field *H* = 0.1 kOe to *f*_1-max_ ≈ 139.5 Hz. The change in frequency under the action of the magnetic field was Δ*f/f*_1_ ≈ 362%. Here Δ*f* = 109.3 Hz is the difference between the highest and the lowest value of the resonance frequency of the structure.

### 3.2. ME Effect Geometry 2 (T-T) 

Similar measurements of the ME effect were carried out in geometry 2. The measured dependences of the ME voltage *u* on the frequency of the AC magnetic field *f* for different values of the bias field *H* are shown in Figure 5. The first bending mode of oscillations is also visible on the frequency dependence of voltage. A significant increase in the maximum amplitude of the ME voltage is observed and constitutes *u*_2_~2200 mV in the field *H*_m2_ = 0.8 kOe, which is larger than the corresponding value obtained for geometry 1 *u*_1_ by the factor of ~58. The peak around 160 Hz observed in the 0.8 kOe field corresponds to the second mode of bending oscillations of the structure.

The dependences of the ME voltage peaks *u*_2_ and the resonance frequency *f*_2_ on the magnetic field *H* were plotted in Figure 6. The voltage at the initial site up to 0.6 kOe steadily increases with the increasing magnetic field, and then there is a sharp increase up to the maximum value *u*_2_ ≈ 2.2 V in the field range from 0.7 kOe to *H*_m2_ = 0.8 kOe, which then monotonically decreases (Figure 6a). The largest calculated MEE coefficient was *ά*_2_ ≈ 134.3 V/(cm∙Oe), which exceeds by 58 times the results reported earlier for stimulation of the structure by magnetic fields in the sample plane due to the larger bend in the PVDF layer.

The field dependence of the resonance frequency for geometry 2 (Figure 6b) also shows a significant difference. In contrast to geometry 1, the frequency changes non-monotonically under the bias magnetic field. The resonance frequency at the starting site decreases from a value of *f*_2_ ≈ 30.9 Hz at *H* = 0.1 kOe to a minimum of *f*_2-min_ ≈ 20.6 Hz in the field *H* = 0.8 kOe, after which it increases again with the increasing field, reaching a value of 76 Hz. The maximum frequency tuning was Δ*f/f*_2min_ ≈ 268.9%, where Δ*f* = 55.4 Hz is the difference between the highest and lowest values of the resonant frequency of the structure.

## 4. Discussion

The difference in the shapes of the field dependences of the ME voltage and resonance frequency at two orientations of the DC and AC magnetic fields is caused by different mechanisms of deformation in the MAE layer.

### 4.1. Influence of the Magnetic Field Direction on the ME Voltage

The dependence of the induced signal on the bias field in the L-T case was found to be non-monotonic with a maximum in the field of 300 Oe (Figure 4a). The MAE layer is stretched in the direction of the bias field due to the magnetodeformation effect. Namely, the particles in the MAE layer displace the elastic matrix in a homogeneous field, as opposed to gradient excitation [33]. The MAE layer was bonded to the substrate, so its stretching, or magnetodeformation, caused it to pull the more elastically hard piezopolymer layer behind it, causing bending. In this case, the stress depends on the first derivative of the magnetodeformation curve of the MAE layer with respect to the magnetic field, similar to the magnetostriction curve for ferromagnetic materials [32].

In the T-T geometry, we consider that the bending of the structure occurs due to the mechanical moment caused by the mechanical instability. Initially, the sample is in a disadvantaged state when its plane is perpendicular to the magnetic field, as mentioned above [37,38]. As published, the magnetization and the magnetic field are noncollinear, leading to the MAE rotation. As the MAE sample is bonded to a stiffer elastic substrate and the composite is fixed at one end, the whole sample bends instead of rotating. The field applied to the sample is amplitude-modulated; therefore, during bending, oscillations of the sample are observed due to the mechanical moment applied to it. The increase in the electric voltage at this stage is due to the increase in the magnetic moment of the MAE sample as the external homogeneous magnetic field is increased, corresponding to the initial part of the field dependence up to 0.7 kOe (Figure 6a).

Furthermore, the state of the sample reaches a kind of mechanical saturation, when the MAE layer becomes almost parallel to the magnetic field. In this position, the MAE exhibits magnetodeformation when the field is further increased, as in L-T geometry. The nonmonotonic dependence of the ME voltage is also observed (as in Figure 4a) in the plot at the field value above 0.8 kOe (Figure 6a). Therefore, in the case of geometry 2 (T-T), the shape of the field dependence is determined by the combined contribution of the mechanical moment generated by the magnetic field and the magnetodeformation of the MAE layer. We assume that the simultaneous contribution of these two mechanisms allows us to describe the shape of the field dependence of the ME voltage, and in the field *H*_m2_ = 0.8 kOe (see Figure 6), the contribution of both mechanisms to the value of the ME effect leads to the maximum ME response of the structure. 

The magnitude of the ME coefficient in geometry 2 exceeds that of geometry 1, which is due to the greater magnitude of the sample deflection in the magnetic field. Table 1 summarizes the previously obtained results in different composite structures based on MAE [30,31,32,33] for comparison. It can be seen that the obtained value of *ά*_1_ ≈ 1.8 V/(cm∙Oe) at magnetization along the long side (geometry L-T) is smaller than that previously obtained for composite structures, which may be due to the non-optimal thickness of the MAE layer [32]. At the same time, the value of the ME coefficient under magnetization perpendicular to the plane of the structure is currently the highest for this type of composite structures, being *ά*_2_ ≈ 134.3 V/(cm∙Oe). To the best of the authors’ knowledge, this value is one of the highest values of the ME coefficient in flexible ME structures. Note also that the increase in the ME coefficient in the T-T geometry can be achieved by increasing the thickness of the MAE layer by increasing the mechanical moment generated by the magnetic field.

Previously in PVDF–amorphous magnetic alloy composite structures in the L-T geometry, values of ME coefficients greater or comparable in magnitude to those obtained in this work have been demonstrated [15,20,22,39,40,41]. Compared to classical structures based on piezoceramics, the order of magnitude of the ME coefficient is in agreement with the results obtained previously [19,42,43,44,45]. The value of the ME coefficients in the T-T geometry for “classical” rigid ME composite structures is smaller than in the L-T configuration due to the stronger influence of demagnetization and the impossibility of bending the PE layer. To the authors’ knowledge, similar studies have not been carried out for flexible structures. The angular dependence of the ME voltage has been measured for an arched structure [44]. 

### 4.2. Influence of the Magnetic Field on the Resonance Frequency

Increasing the bias field leads to an increase in the resonant frequency in the L-T geometry (Figure 4b). The dependence of the resonant frequency on the bias field in the T-T geometry can be divided into two parts. The first is the gradual bending of the sample, which is described by the frequency decay (Figure 6b). As the field continues to increase, the other mechanism—MAE magnetodeformation—dominates, and the frequency increases, as in the L-T case. The discussion of this is presented below.

As shown above, the resonant frequency tuning under the influence of the magnetic field was Δ*f/f*_1_ = 362% and Δ*f/f*_2_ = 268.9% in geometries 1 and 2, respectively. In the same 2 kOe fields, the resonance frequency tuning was Δ*f*_2_*/f*_min_ ≈ 179.1% (Δ*f*_2_ = 57.5 Hz) in the T-T geometry, which is about 2.5 times smaller than in the L-T geometry. It should be noted that, to the best of our knowledge, the minimum resonance frequency of the structure *f*_min_ ≈ 20.6 Hz is the lowest among the ME composite structures. Previously, the lowest value was approximately 27 Hz and was observed for the PVDF–Metglas structure [21]. The frequency tuning values obtained are much larger than those for hematite-based structures (in saturation Δ*f/f* = 24%) [34] and larger than those for similar PVDF–MAE structures in geometry 1 in the magnetic field *H* = 3 kOe (Δ*f/f* = 216%) [32].

### 4.3. Calculation of the Resonant Frequency in Magnetic Field

The effect of the magnetic field on the frequency change in geometry 1 (L-T) was analytically estimated. The magnetic field induces a mechanical moment, which can be written as ${\vec{m}}_{mm}=\vec{B}\times {\vec{M}}_{magn}$, where $\vec{B}$ is the magnetic induction and ${\vec{M}}_{magn}$ is the magnetization of the MAE layer. This moment is associated with the bending of the structure (similar to beam bending) and tends to return the structure to the equilibrium state. It should also be noted that we do not take into account the displacement of the filler particles in the MAE layer and the corresponding magnetorheological effect. Then, the equations of the balance of the acting forces (1) and moments (2) have the following forms: (1)$$-\frac{\partial F}{\partial x}-\rho Sdx\frac{\partial^{2}y}{\partial t^{2}}=0$$
(2)$$M-M-\frac{\partial M}{\partial x}dx+Fdx+BM_{magn}S_{m}dx\frac{\partial y}{\partial x}=0$$
where y is the displacement of the sample points; F is the vertical force; $\rho Sdx\frac{\partial^{2}y}{\partial t^{2}}$ is the inertia force; ρ is the effective density of the structure, which is equal to $\rho =\frac{\rho_{MAE}t_{MAE}+\rho_{PEP}t_{PEP}}{t_{MAE}+t_{PEP}}$; $\rho_{MAE}$, $t_{MAE}$, $\rho_{PEP}$, and $t_{PEP}$ are density and thickness of MAE and piezopolymer, respectively; S is the cross-sectional area of the sample; ρSdx is the mass of the sample; $\frac{\partial^{2}y}{\partial t^{2}}$ is the acceleration. M and $M+\frac{\partial M}{\partial x}dx$ are the pair of moments occurring from samples bending; $BM_{magn}S_{m}dx\frac{\partial y}{\partial x}$ is the mechanical torque generated by the magnetic field; $M_{magn}$ is the magnetization of the MAE layer; $S_{m}dx$ is the MAE volume; $S_{m}$ is the MAE cross-sectional area; $\frac{\partial y}{\partial x}$ is the angle between the magnetic field direction and the magnetic moment of the MAE layer. The mechanical torque equation includes the cross-sectional area of the MAE layer because the magnetic field only affects the magnetic layer of the structure.

The expression for the mechanical moment of the two-layer structure can be written in the following form [46]:(3)$$M=\left(E_{PEP}I_{PEP}+E_{MAE}I_{MAE}\right)\frac{\partial^{2}y}{\partial x^{2}}$$
where $E_{PEP}$, $I_{PEP}$, $E_{MAE}$, and $I_{MAE}$ are Young’s modulus and the moment of inertia of the PEP layer and MAE layer, respectively. Substituting the force F in Equation (2) and substituting it into Equation (1), taking into account Equation (3), one can obtain:(4)$$\left(E_{PEP}I_{PEP}+E_{MAE}I_{MAE}\right)\frac{\partial^{4}y}{\partial x^{4}}+BM_{magn}S_{m}\frac{\partial^{2}y}{\partial x^{2}}=-\rho S\frac{\partial^{2}y}{\partial t^{2}}$$

The solution of Equation (4) for the coordinates of each point of the structure at the natural oscillation frequency can be written in the following form:(5)$$y=\zeta \left(\xi_{1}\cdot \cos\omega t+\xi_{2}\cdot \sin\omega t\right)$$
where ω is the circular oscillation frequency. Function ζ specifies the oscillation shape of the beam, and $\xi_{1}$, $\xi_{2}$ are the oscillation amplitudes. Equation (5) has been substituted into Equation (4) using the known boundary conditions for the cantilever of length *L* [27,47], as follows:(6)$$\left\{\begin{array}{c} \zeta \left(0\right)=0\\ \zeta \prime \left(0\right)=0\\ \zeta \prime \prime \prime \left(L\right)=0\\ \zeta^{\prime \prime \prime \prime }\left(L\right)=0 \end{array}\right.$$

In the case of no magnetic field *B* = 0, Equation (4) is reduced to the known form [27,47] for the frequencies of the bending oscillations of the cantilever and has the following solution:(7)$$f_{0}=\frac{k^{2}}{2\pi L^{2}}\sqrt{\frac{E_{p}I_{p}+E_{m}I_{m}}{\rho S}}$$
where k ≈ 1.875 is a constant and L is the length of the cantilever. Calculation using Equation (7) allows us to obtain the value of the resonance frequency *f*_0_ ≈ 31.9 Hz, which differs from the experimental one by ~6%, giving a good agreement between theory and experiment.

The dependence of the resonance frequency *f*_1_ on the magnetic field was calculated by numerical methods in the Mathcad15 software (PTC, Burlington, ON, Canada). The known material parameters L = 19.6 mm, *t*_PEP_= 220 μm, *E*_m_ ≈ 90 kPa, *E*_PEP_ ≈ 4.9 GPa, *ρ*_m_≈ 3090 kg/m^3^, and *ρ*_PEP_ ≈ 1140 kg/m^3^, as well as the values of magnetizations and magnetic fields taken from the magnetic hysteresis loop of the MAE layer shown in Figure 7 were substituted into Equation (4). The resulting theoretical curve is plotted as a solid line in Figure 4b. It can be seen that the dependence qualitatively describes the frequency growth under the external bias magnetic field. The predicted value of the resonant frequency of 82.7 Hz in the field *H* = 2 kOe is about 1.7 times smaller than the experimental one. Estimation of the resonant frequency tuning gives the value of *f/f*_min_ ≈ 159.2%, which is 2.3 times smaller than the experimental one. For a more accurate estimation, it is necessary to take into account not only the influence of the returning mechanical moment resulting from the magnetic field, but also the influence of the forces acting on the piezopolymer layer from the MAE side during its stretching, such as the magnetorheological effect, as well as the influence of the rotational inertia and the weight of the beam.

It is important to point out that when the magnetic field causes the structure to deviate from its equilibrium position in geometry 2, the field term in Equation (2) is expressed with a negative sign. Then, based on the solution of Equation (4), it is determined that the frequency of the first mode decreases. This decrease is clearly seen in the first plot shown in Figure 6b, where it is represented by the red dashed line. It can be seen that the dependence is well described at the initial site when the deflection angle is minimal. When the deflection angle of the structure exceeds a certain threshold, the proposed approach becomes ineffective as the oscillations deviate significantly from being small. In geometry 2, the combined effect of the MAE layer deformation and the deviation from the equilibrium position is believed to cause a non-monotonic frequency change under the magnetic field. Unfortunately, this theory does not allow us to predict the overbending of the structure in the magnetic field, but it does allow us to qualitatively predict the sign of the change in the resonance frequency. Once a certain level of mechanical saturation is reached, the sample begins to oscillate in accordance with the mechanism outlined for the first geometry. Due to the unknown length of the cantilever, the calculations did not yield precise quantitative results. The second part of the plot exhibits a qualitative feature of frequency increasing with an increasing bias field, similar to that shown in Figure 4b.

## 5. Conclusions

In this work, the resonant magnetoelectric effects in a flexible PVDF–MAE composite structure in uniform AC and DC magnetic fields in two geometries were investigated for the first time. The maximum value of the ME coefficient was 134.3 V/(cm∙Oe), which is in the order of magnitude of the largest results for flexible composite multiferroics. This value was two orders of magnitude larger for the perpendicular application of AC and DC magnetic fields to the plane of the layered sample than that in parallel application. In such a geometry, one of the fascinating results was the non-monotonic behavior of the field dependence of resonant frequency. This was explained by the different mechanisms of bending of the layered structure, which are associated with mechanical instability and magnetodeformational effect. The largest obtained frequency tuning of the ME structure under the magnetic field was 362% at the bias field increase of 2 kOe, which is the maximum value among available results for multiferroics. The lowest resonance frequency for the composite structure was also obtained and was ~20.6 Hz. The results can be of use for the development of low-frequency devices for signal receiving and processing, magnetic field sensors, and energy harvesting devices. By varying the composition of the structure, the frequency and field range can be tuned for tunable applications. One of the essential and challenging technological issues in the introduction of composites into practical applications is the potential for mass production technologies.

## Figures and Tables

**Figure 1 polymers-16-00928-f001:**
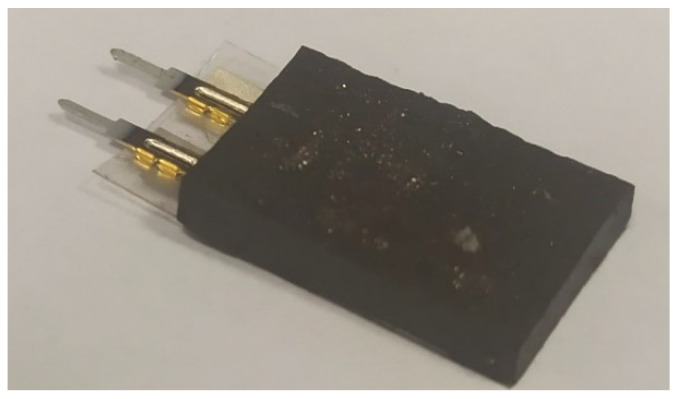
Photograph of the PEP-MAE sample.

**Figure 2 polymers-16-00928-f002:**
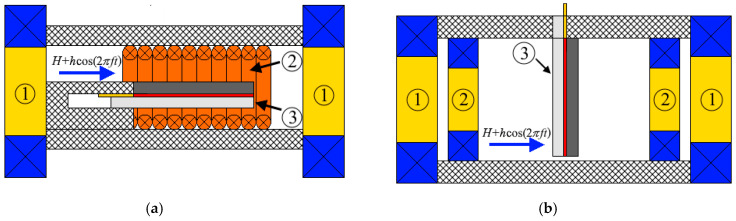
(**a**) Schematic representation of the measuring set-up in geometry 1. (**b**) Schematic representation of the measuring set-up in geometry 2. 1—poles of the electromagnet, 2—excitation coils, 3—sample.

**Figure 3 polymers-16-00928-f003:**
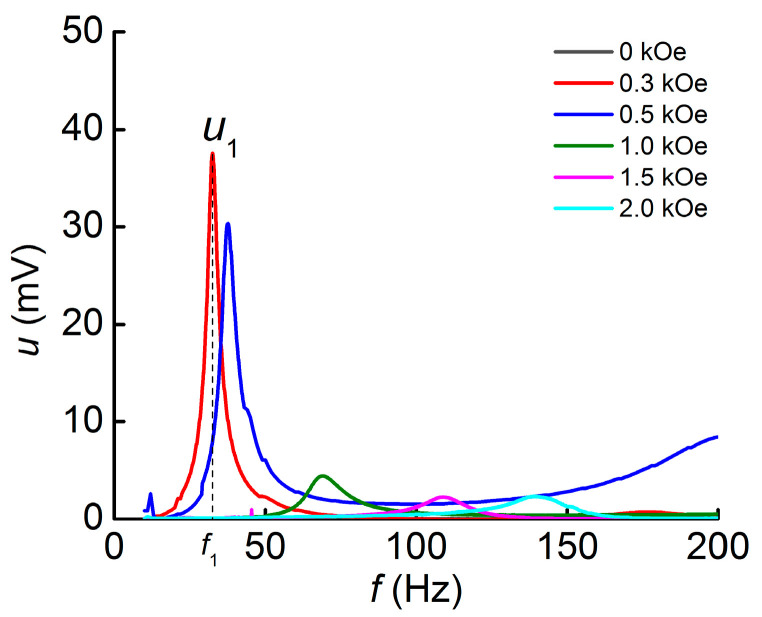
Dependences of the induced voltage of the structure *u* on the frequency of the alternating magnetic field *f* measured at different values of the bias field for geometry 1.

**Figure 4 polymers-16-00928-f004:**
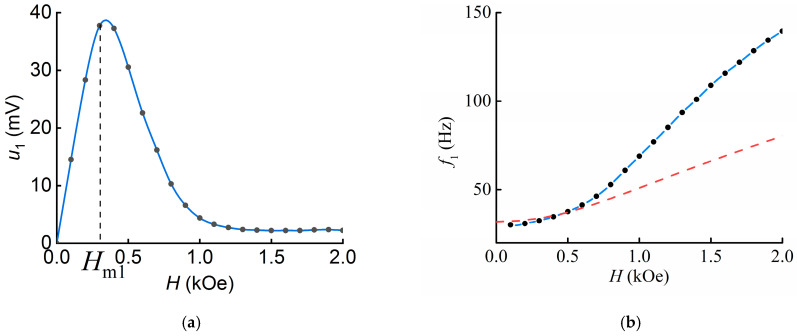
Dependence of (**a**) induced voltage *u*_1_ at the resonance frequency and (**b**) resonance frequency *f*_1_ on the bias magnetic field *H* in geometry 1. Dots are experimental results, the blue curve is the spline, and the red curve is the calculation.

**Figure 5 polymers-16-00928-f005:**
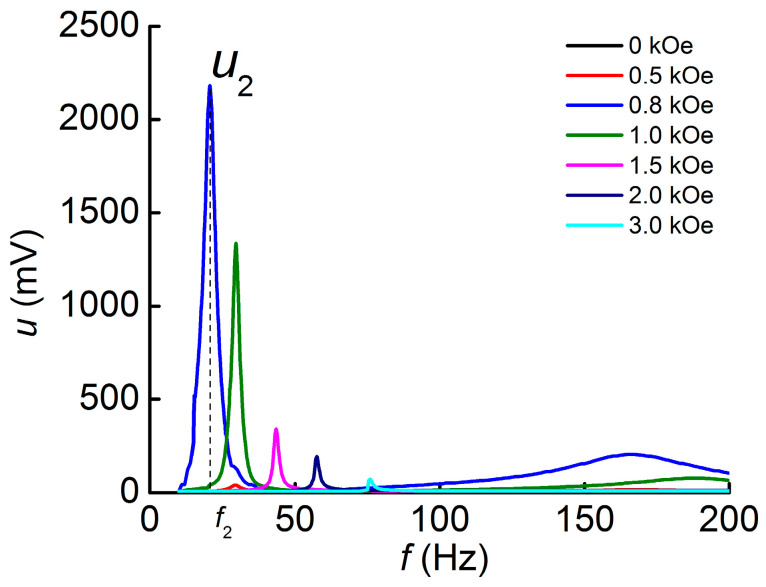
Dependences of the ME voltage *u* on the frequency of the AC magnetic field *f* measured at different values of the bias field *H* in geometry 2.

**Figure 6 polymers-16-00928-f006:**
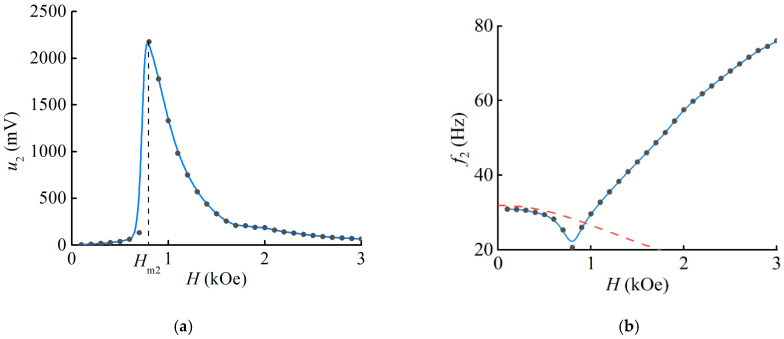
Dependence of (**a**) ME voltage *u*_2_ at the resonance frequency and (**b**) resonance frequency *f*_2_ on the bias magnetic field in geometry 2. The dots are experimental results, the blue curves are the spline, and the red curve is the calculation.

**Figure 7 polymers-16-00928-f007:**
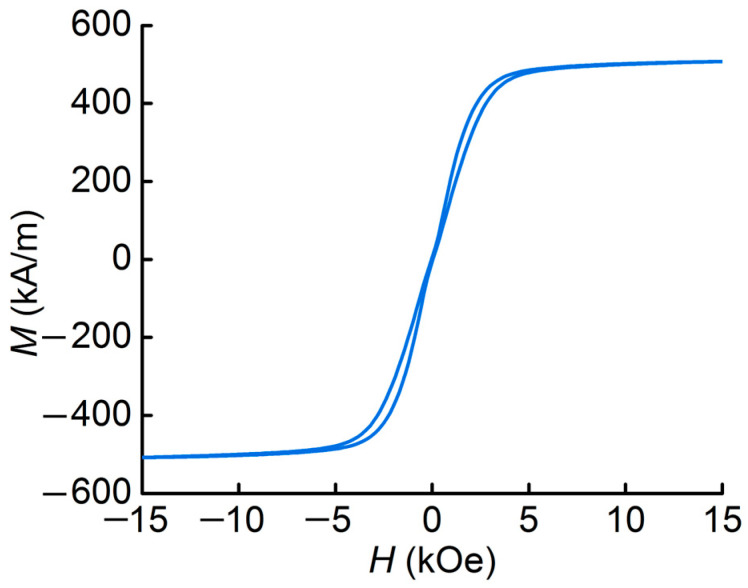
Hysteresis loop for the MAE sample.

**Table 1 polymers-16-00928-t001:** Values of ME coefficients in PVDF-MAE composite structures.

Measurement Geometry	ME Coefficient *ά*, V/(cm∙Oe)	Reference
L-T	6.4	[32]
T-T	50	[30]
T-T	79	[31]
T-T	0.77	[33]
L-T	1.8	This work
T-T	134.3	This work
T-T	0.19	[25]

## Data Availability

Data are contained within the article.

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
