# Peer review of "Low-Frequency Resonant Magnetoelectric Effect in a Piezopolymer-Magnetoactive Elastomer Layered Structure at Different Magnetization Geometries"

_polymers, 2024, doi:10.3390/polym16070928_

Round 1
Reviewer 1 Report
Comments and Suggestions for Authors
Well written. Following can be improved :
Add calculation curve in fig 6b for geoemetry 2 this will help the reader understand the non-monotonic behavior here better.
Would the geometry of the setup play a role on the hysterisis loop of the MAE sample , if so why ?
Author Response
We thank the reviewers for their careful attention to the manuscript and for the comments indicated. We are confident that the corrections made in accordance with the reviewers' comments have significantly improved the quality of our work.
1.1. Add calculation curve in fig 6b for geometry 2 this will help the reader understand the non-monotonic behavior here better.
Answer. We carried out the calculations in geometry 2. “It can be seen that the dependence is well described at the initial site when the deflection angle is minimal. When the deflection angle of the structure exceeds a certain threshold, the proposed approach becomes ineffective as the oscillations deviate significantly from being small. In geometry 2, the combined effect of the MAE layer deformation and the deviation from the equilibrium position is believed to cause a non-monotonic frequency change under the magnetic field. Unfortunately, this theory does not allow us to predict the overbending of the structure in the magnetic field, but it does allow to qualitatively predict the sign of the change in the resonance frequency. Once a certain level of mechanical saturation is reached, the sample begins to oscillate in accordance with the mechanism outlined for the first geometry. Due to the unknown length of the cantilever, the calculations did not yield precise quantitative results. The second part of the plot exhibits a qualitative feature of frequency increasing with an increasing bias field, similar to that shown in Figure 4b.”
(please see the attachment)
1.2. Would the geometry of the setup play a role on the hysteresis loop of the MAE sample , if so why ?
Answer. The magnetic hysteresis loop was measured for a 4*4*1 mm sample (rectangular form), with the external magnetic field directed parallel to the sample plane. For measurements in the perpendicular direction, the demagnetising factor must be taken into account during processing, as well as the geometrical factor, due to which the distance from the sample plane to the receiving coils increases. In our case, we specifically carried out the magnetic measurements on the VSM so that the results were obtained for the material, but not for the sample.
The hysteresis loop of the dependence of the induced voltage on the bias magnetic field could also occur due to the hysteresis dependence of the deformation and elastic modulus of the magnetic elastomer on the magnetic field and the shape memory effect. In our case, the maximum value of the applied external field was 2 kOe, which is less than the saturation field. Therefore, the hysteresis magnetoelectric effect was not investigated.

Reviewer 2 Report
Comments and Suggestions for Authors
Reviewer Comments:
The manuscript entitled “Low-frequency resonant magnetoelectric effect in piezopolymer magnetoactive elastomer layered structure at different magnetization geometries” by Savelev et al have focussed on measuring Low-frequency resonant magnetoelectric effect in piezopolymer - magnetoactive elastomer and he found a giant value frequency tuning in a magnetic field of up to 362% demonstrated so as one of the largest values of the magnetoelectric effect observed in polymer multiferroics, up to 134.3 V/(Oe∙cm). The interaction mechanism that could further support the study should be provided by the author. Result and dissuasion part is well explained and supported with relevant reference. It is suggested that they address the following points before the acceptance.
Ø In Abstract, Author should provide that focus for key problems, material preparation, characterization, and key result with concluding remarks.
Ø In the last paragraph of introduction, the advantages/superiorities of the present study are needed to be discussed by comparing with some literature examples.
Ø content of the manuscript
Ø Author should also incorporate the thermal analysis of prepared s MAE sample by TGA and DSC
Ø The author should also more discussion about the Hysteresis loop for MAE sample with respect to temperature
Ø Conclusions of the manuscript should be more focused and presented point wise along with future significance
Ø Provide the cat log number of the chemical porches in sigma and somewhere else
Ø throughout the manuscript: separate the numbers from the units (i.e.: 5 mg instead of 5mg)
Ø Author may also incorporate the surface morphology study of prepared MAE & PEP-MAE samples
Comments on the Quality of English LanguageExtensive editing of English language required
Author Response
We thank the reviewers for their careful attention to the manuscript and for the comments indicated. We are confident that the corrections made in accordance with the reviewers' comments have significantly improved the quality of our work.
The manuscript entitled “Low-frequency resonant magnetoelectric effect in piezopolymer magnetoactive elastomer layered structure at different magnetization geometries” by Savelev et al have focused on measuring Low-frequency resonant magnetoelectric effect in piezopolymer - magnetoactive elastomer and he found a giant value frequency tuning in a magnetic field of up to 362% demonstrated so as one of the largest values of the magnetoelectric effect observed in polymer multiferroics, up to 134.3 V/(Oe∙cm). The interaction mechanism that could further support the study should be provided by the author. Result and discussion part is well explained and supported with relevant reference. It is suggested that they address the following points before the acceptance.
2.0. The interaction mechanism that could further support the study should be provided by the author.
Physically, the process of interaction of the sample with the magnetic field is described in the text:
In the geometry L-T, a uniform magnetic field acts on a magnetic elastomer, and a magnetodeformation effect is observed, i.e. stretching or compressing of the elastomer layer. Since it is bonded to a stiffer substrate, the whole sample bends.
In the geometry T-T, the magnetic elastomer sample has mechanical instability due to the magnetic field being directed perpendicular to the plane of the MAE sample. The sample tends to rotate so that the magnetic field is directed along the long side towards the plane.
To build the model we need the result of the sample bending, not its first-principles mechanisms. Therefore, the mechanical moment ,which is necessary to formulate the basic equation of bending vibrations of a rectangular beam, is taken into account.
In each case, both (constant and alternating) applied fields are homogeneous.
2.1. In Abstract, Author should provide that focus for key problems, material preparation, characterization, and key result with concluding remarks.
Answer.
The section Abstract was corrected and included required points.
“The search for novel materials with enhanced characteristics for the advancement of flexible electronic devices and energy harvesting devices is currently a significant concern. Multiferroics are a prominent example of energy conversion materials. The magnetoelectric conversion in a flexible composite based on a piezopolymer layer and a magnetic elastomer layer was investigated. The study focused on investigating the dynamic magnetoelectric effect in various configurations of external alternating and constant homogeneous magnetic fields (L-T and T-T configurations). The T-T geometry exhibited a two orders of magnitude higher coefficient of the magnetoelectric effect compared to the L-T geometry. Mechanisms of structure bending in both geometries were proposed and discussed. A theory was put out to explain the change of the resonance frequency in a uniform external field. A giant value of frequency tuning in a magnetic field of up to 362 % was demonstrated; one of the highest values of the magnetoelectric effect yet recorded in polymer multiferroics was observed, reaching up to 134.3 V/(Oe∙cm).”
2.2. In the last paragraph of introduction, the advantages/superiorities of the present study are needed to be discussed by comparing with some literature examples.
Answer. The comparing with literature data was carried out and presented in the Introduction section and in the section Results and Discussions. In the last paragraph of Introduction the advantages and peculiarities of presented studies was formulated. Corresponding changes have been made to the text.
“However, the flexibility, stretchability and foldability are limited for these materials that can impair the effectiveness and service life of the entirely magnetoelectric material. … Flexible magnetoelectric elements are based on the polymer components that allows to extent their incorporation into technological device applications such as flexible displays, soft tools, magnetoactive robotics, biomedical applications. … Flexible structures have low values of Young's moduli, which makes it possible to reduce their frequencies of bending vibrations to tens of Hz - units of kHz. … The study of the controlled resonant frequency tuning via the ME effect across a wide frequency range in flexible composite structures is of significant interest for subsequent practical applications, especially for ME tunable low-frequency antennas and sensors. … The work presents the results of the resonance magnetoelectric effect analysis in bilayer structure “piezopolymer PVDF – magnetic elastomer” (PEP-MAE-Fe77) with the 3 mm MAE layer thickness, which contains iron microparticles with mass fraction 77 mass %. Young’s modulus of MAE composite was 90 kPa, that allowed to investigate the magnetoelectric effect at low frequencies. … The maximum value of the ME effect obtained was 134.3 V/(cm∙Oe), which is one of the highest values among two-layered PVDF-based multiferroics. The giant frequency shift of ~362 % was observed in the bias magnetic field and it is the largest value among existing results. Such a flexible magnetoelectric element allows to construct flexible technological devices without leakage currents, with remote access and good stretchability and foldability.”
2.3. content of the manuscript
Answer. Content of the manuscript was improved according to the reviewers recommendations.
2.4. Author should also incorporate the thermal analysis of prepared s MAE sample by TGA and DSC
Answer. All measurements were carried out at room temperature, the temperature did not change during the measurements. Therefore, the temperature analysis of phase transitions is not given in the paper. However, such an analysis was carried out and showed that:
The process of thermal decomposition of silicone polymer starts at about 300 C, the first exothermal peak is formed at 345 C. After reaching the first exothermal peak, no additional peaks were detected on the DSC curve during subsequent heating and cooling cycles, which indicates the temperature stability of silicone in this temperature range. When the temperature decreases from room temperature, the process of glass transition of the polymer occurs at temperatures in the range of -76 C - -36 C, i.e. its solidification without the formation of a crystalline structure. Thus, no phase transition occurs at room temperature, no changes in the MEE are observed in this temperature region.
2.5. The author should also more discussion about the Hysteresis loop for MAE sample with respect to temperature
Answer.
Since there are no temperature studies in the paper, there is no temperature analysis of the hyste
resis loop of the magnetic elastomer.
At the same time, if we refer to previous works with the presence of such a temperature study, it can be seen that the hysteresis loop of MAE changes its course when the temperature drops below the glass transition temperature. For example, two hysteresis loops are presented at the Figure below, which were measured at room temperature and at the 100 K.
References for these works are:
Bodnaruk, A. V., Brunhuber, A., Kalita, V. M., Kulyk, M. M., Snarskii, A. A., Lozenko, A. F., … Shamonin, M. (2018). Temperature-dependent magnetic properties of a magnetoactive elastomer: Immobilization of the soft-magnetic filler. Journal of Applied Physics, 123(11), 115118. doi:10.1063/1.5023891.
Kiarie, W. M., Gandha, K., & Jiles, D. C. (2021). Temperature-Dependent Magnetic Properties of Magnetorheological Elastomers. IEEE Transactions on Magnetics, 1–1. doi:10.1109/tmag.2021.3082302.
Bodnaruk, A. V., Kalita, V. M., Kulyk, M. M., Lozenko, A. F., Ryabchenko, S. M., Snarskii, A. A., … Shamonin, M. (2018). Temperature blocking and magnetization of magnetoactive elastomers. Journal of Magnetism and Magnetic Materials. doi:10.1016/j.jmmm.2018.10.0.
2.6. Conclusions of the manuscript should be more focused and presented point wise along with future significance
The conclusions were supplemented by more specific results obtained.
“In this work, the resonant magnetoelectric effects in a flexible PVDF-MAE composite structure in uniform AC and DC magnetic fields in two geometries were investigated for the first time. The maximum value of the ME coefficient was 134.3 V/(cm∙Oe), which is in the order of magnitude of the largest results for flexible composite multiferroics. This value was two orders of magnitude larger for the perpendicular application of AC and DC magnetic fields to the plane of the layered sample than that in parallel application. In such a geometry one of the fascinating results was the non-monotonic behavior of the field dependence of resonant frequency. This was explained by different mechanisms of bending of the layered structure, which are associated with mechanical instability and magnetodeformational effect. The largest obtained frequency tuning of the ME structure under the magnetic field was 362 % at the bias field increase of 2 kOe, that is the maximum value among available results for multiferroics. The lowest resonance frequency for the composite structure was also obtained and was ~20.6 Hz. The results can be of use for the development of low-frequency devices for signal receiving and processing, magnetic field sensors, and energy harvesting devices. By varying the composition of the structure, the frequency and field range can be tuned for tunable applications. One of the essential and challenging technological issues in the introduction of composites into practical applications is the potential for mass production technologies.”
2.7. Provide the catalog number of the chemical porches in sigma and somewhere else
Answer.
In this paper, we use a previously fabricated magnetic elastomer composite. The SIEL composite is commercially available, but we have not found its classification in catalogues. Below are several papers that describe the methodology and materials for fabricating magnetic elastomers with iron particles.
Development of magnetoactive elastomers for sealing eye retina detachments / Y. A. Alekhina, L. A. Makarova, S. A. Kostrov et al. // Journal of Applied Polymer Science. — 2019. — Vol. 136, no. 17. — P. 47425–47425(9).
New composite elastomers with giant magnetic response / A. V. Chertovich, G. V. Stepanov, E. Y. Kramarenko, A. R. Khokhlov // Macromolecular Materials and Engineering. — 2010. — Vol. 295, no. 4. — P. 336–341.
Alekseeva, E.I., Nanush’yan, S.R., Ruskol, I.Y. et al. Silicone compounds and sealants and their application in various branches of industry. Polym. Sci. Ser. D 3, 244–248 (2010). https://doi.org/10.1134/S1995421210040076
2.8. throughout the manuscript: separate the numbers from the units (i.e.: 5 mg instead of 5mg)
Answer. Thanks to the reviewer and we have had a look at it.
2.9. Author may also incorporate the surface morphology study of prepared MAE & PEP-MAE samples
Answer.
The surface morphology of the composite was not investigated. In this case, PVDF layer and MAE layer are glued together with silicone sealant. The thickness of the MAE layer is 3 mm. The magnetoelectric effect is observed when a volumetric magnetic field is applied to the MAE layer. When bending, stresses and strains will be observed on the MAE surface. With such an inseparable bonding of the two layers, the surface properties of MAE will possibly change compared to a MAE. However, such changes cannot affect the magnitude of the MEE, since its magnitude is measured by the voltage generated during bending of the piezoelectric polymer and not by the magnitude of the composite deformation.
Please see the attachment.

Reviewer 3 Report
Comments and Suggestions for Authors
1. The numbering of images throughout the text should be harmonized. For example, Fig. 2 on line 163 and Figure 2 on line 167.
2. In calculating the forces (1) and (2), it is assumed that the MAE layer is uniformly magnetized in order to simplify the calculations. How much do the results obtained by this method of calculation deviate from reality?
3. The authors should further elaborate on why the bending of the structure is caused by the inhomogeneous magnetization of the sample?
4. The author's description of the ME factor as one of the highest values in the text is not convincing without the support of the referenced article. Please compare peer work to highlight the advancement of this work.
5. In his paper, the author analyzes the deformation of the structure and argues that it is produced by an inhomogeneous magnetic field. However, in the calculation of moments, the magnetic field is considered to be uniform. Please clarify this statement and verify it with reasonable calculations.
Author Response
We thank the reviewers for their careful attention to the manuscript and for the comments indicated. We are confident that the corrections made in accordance with the reviewers' comments have significantly improved the quality of our work.
3.1. The numbering of images throughout the text should be harmonized. For example, Fig. 2 on line 163 and Figure 2 on line 167.
Answer. Done.
3.2. In calculating the forces (1) and (2), it is assumed that the MAE layer is uniformly magnetized in order to simplify the calculations. How much do the results obtained by this method of calculation deviate from reality?
Answer.
When building the model, the goal was to qualitatively estimate the change of the resonant frequency of the structure in an external displacement field. To determine the mechanical torque that acts on the sample, there is no need to introduce any inhomogeneous magnetisation distribution over the sample. Initially, the external magnetic field is homogeneous, the internal magnetic field is nearly homogeneous considering demagnetisation, and the magnetisation distribution is homogeneous. The difference in the results between theory and experiment at small oscillations is not more than 10%. Further, the results do not coincide quantitatively, since the model must take into account the change in the elastic modulus of the elastomer in the magnetic field, the mass of the cantilever, the inhomogeneity of the magnetic field, the inhomogeneity of the magnetisation distribution of the sample in the bent state, etc.
3.3. The authors should further elaborate on why the bending of the structure is caused by the inhomogeneous magnetization of the sample?
Answer.
We have for the moment dispensed with the need to mention the inhomogeneity of the magnetic field. We thank the reviewer for drawing attention to this point.
The answer to this question is presented in more detail in (5). Nevertheless, inhomogeneity of the magnetisation distribution can occur in a sample, and the mechanism is further described.
The inhomogeneity of the magnetic field in T-T geometry arises from the inhomogeneity of the demagnetising field distribution. The demagnetising field at each point of the sample is directed perpendicular to the surface at that point. Accordingly, the internal field in the sample is inhomogeneously distributed, resulting in inhomogeneous magnetisation of the sample. In this model, we do not account for the inhomogeneity. It is likely that such inhomogeneity will increase the mechanical moment, which will lead to refinement of theoretical calculations.
3.4. The author's description of the ME factor as one of the highest values in the text is not convincing without the support of the referenced article. Please compare peer work to highlight the advancement of this work.
Answer. A comparison of the results is given in the article in the Introduction as well as in the description of the results:
“The layered structures and the composites where the ferromagnetic particles embedded inside polymer PVDF are often employed for investigations of ME effect. The ME coefficient in such structures can reach significant values of several V/(cm∙Oe) [15,18-21]. These composites can be prepared of different shapes and geometries and can be biocompatible. One of the highest values of the ME coefficient was established for PVDF-TrFE/Metglas laminated structure (850 V/(cm∙Oe)) [21]…
Table 1 summarises the previously obtained results in different composite structures based on MAE [31-34] for comparison. It can be seen that the obtained value of ά1 ≈ 1.8 V/(cm∙Oe) at magnetization along the long side (geometry L-T) is smaller than that previously obtained for composite structures, which may be due to the non-optimal thickness of the MAE layer [33]. At the same time, the value of the ME coefficient under magnetisation perpendicular to the plane of the structure is currently the highest for this type of composite structures, being ά2 ≈ 134.3 V/(cm∙Oe). To the best of the authors' knowledge, this value is one of the highest values of the ME coefficient in flexible ME structures. …
Previously, in PVDF-amorphous magnetic alloy composite structures in the L-T geometry, values of ME coefficients greater or comparable in magnitude to those obtained in this work have been demonstrated [15,20,22,41-43]. Compared to classical structures based on piezoceramics, the order of magnitude of the ME coefficient is in agreement with the results obtained previously [19,44,45]. The value of the ME coefficients in the T-T geometry for "classical" rigid ME composite structures is smaller than in the L-T configuration due to the stronger influence of demagnetisation and the impossibility of bending the PE layer. To the authors' knowledge, similar studies have not been carried out for flexible structures. Angular dependence of the ME voltage has been measured for an arched structure [46].”
3.5. In this paper, the author analyzes the deformation of the structure and argues that it is produced by an inhomogeneous magnetic field. However, in the calculation of moments, the magnetic field is considered to be uniform. Please clarify this statement and verify it with reasonable calculations.
Answer.
We thank the reviewer for this indication and believe that in the first version of the manuscript the cause of sample deformation was not well formulated. To build the model we need the result of the sample bending, not its first-principles mechanisms. Therefore, the mechanical moment ,which is necessary to formulate the basic equation of bending vibrations of a rectangular beam, is taken into account.
In each case, both (constant and alternating) applied fields are homogeneous. We have removed the typo about the inhomogeneity of the external field.
Geometry 1. The MAE layer is stretched in the direction of the bias field due to magnetodeformation effect. Namely, the particles in MAE layer displace in elastic matrix in homogeneous field. The MAE layer was bonded to the substrate, so its stretching, or magnetodeformation, causes it to pull the more elastically hard piezopolymer layer behind it, that causes the bending. In this case, the stress depends on the first derivative of the magnetodeformation curve of the MAE layer on the magnetic field similarly to the magnetostriction curve for ferromagnetic materials [33].
Geometry 2 (T-T). “In the T-T geometry, we consider that the bending of the structure occurs due to the mechanical moment caused by the mechanical instability. Initially, the sample is in a disadvantaged state when its plane is perpendicular to the magnetic field, as it was mentioned above [38, 39]. As published, the magnetization and the magnetic field are noncollinear, leading to the MAE rotation. As the MAE sample is bonded to a stiffer elastic substrate and the composite is fixed at one end, the whole sample bends instead rotating. The field applied to the sample is amplitude-modulated; therefore, during bending, oscillations of the sample are observed due to the mechanical moment applied to it.
The increase in the electric voltage at this stage is due to the increase in the magnetic moment of the MAE sample as the external homogeneous magnetic field is increased, corresponding to the initial part of the field dependence up to 0.7 kOe (Figure 6a).”
We carried out the calculations in geometry 2. “It can be seen that the dependence is well described at the initial site when the deflection angle is minimal. When the deflection angle of the structure exceeds a certain threshold, the proposed approach becomes ineffective as the oscillations deviate significantly from being small. In geometry 2, the combined effect of the MAE layer deformation and the deviation from the equilibrium position is believed to cause a non-monotonic frequency change under the magnetic field. Unfortunately, this theory does not allow us to predict the overbending of the structure in the magnetic field, but it does allow to qualitatively predict the sign of the change in the resonance frequency. Once a certain level of mechanical saturation is reached, the sample begins to oscillate in accordance with the mechanism outlined for the first geometry. Due to the unknown length of the cantilever, the calculations did not yield precise quantitative results. The second part of the plot exhibits a qualitative feature of frequency increasing with an increasing bias field, similar to that shown in Figure 4b.”

Round 2
Reviewer 3 Report
Comments and Suggestions for Authors
It has been fully modified and can be accepted.